# Relationship between exclusive breastfeeding and brain-derived neurotrophic factor in children

**Carlos Berlanga-Macías**[1][◉], **Mairena Sánchez-López**[1,2][◉], **Montserrat Solera-Martínez**[1][◉]*, **Ana Díez-Fernández**[1][◉], **Inmaculada Ballesteros-Yáñez**[3][◉], **Carlos A. Castillo-Sarmiento**[3][◉], **Isabel A. Martínez-Ortega**[1][◉], **Vicente Martínez-Vizcaíno**[1,4][◉]

**1** Social and Health Care Research Center, Universidad de Castilla-La Mancha, Cuenca, Spain, **2** Faculty of Education, Universidad de Castilla-La Mancha, Ciudad Real, Spain, **3** Faculty of Medicine, Universidad de Castilla-La Mancha, Ciudad Real, Spain, **4** Faculty of Health Sciences, Universidad Autónoma de Chile, Talca, Chile

◉ These authors contributed equally to this work.
* Montserrat.Solera@uclm.es

**Data Availability Statement:** All relevant data are within the manuscript and its Supporting Information files.

## Abstract

### Objective

A positive relationship between breastfeeding and brain-derived neurotrophic factor (BDNF) in infants has been suggested due to the presence of BDNF in human milk. This study aimed to determine the relationship between exclusive breastfeeding and BDNF serum levels in Spanish schoolchildren.

### Methods

A cross-sectional analysis including 202 schoolchildren, aged eight to 11 years, from Cuenca, Spain, was conducted. Information on sociodemographic and anthropometric variables, sexual maturation, birth weight and exclusive breastfeeding ('no exclusive breastfeeding', and exclusive breastfeeding for ≤6 and >6 months), and BDNF serum levels using an ELISA method were obtained. Covariance analyses (ANCOVA) were conducted to examine the relationship between serological BDNF and exclusive breastfeeding after controlling for potential confounders.

### Results

ANCOVA models showed no significant differences in BDNF levels between children who were exclusively breastfed for more than six months versus those who were not (p > 0.05). No significant differences were observed by age group (eight to nine years versus 10 to 11 years; p > 0.05). Additionally, no clear negative trend in BDNF serum levels according to sexual maturation categories was found (p > 0.05).

### Conclusion

These findings suggest that exclusive breastfeeding does not have a significant positive association on BDNF from eight to 11 years, since children who were exclusively breastfed

**Funding:** This research is based on data from a cluster randomized controlled trial (registration number NCT03236337), which has been funded by the Spanish Ministry of Economy and Competitiveness - Carlos III Health Institute (Fondo de Investigación en Salud [FIS] PI16/01919). The funders had no role in study design, data collection and analysis, decision to publish, or preparation of the manuscript.

**Competing interests:** The authors have declared that no competing interests exist.

did not have significantly higher BDNF levels than those who were not exclusively breastfed. Likewise, BDNF levels were not found to be negatively affected by hormonal development. Future research should examine the influence of exclusive breastfeeding on BDNF over the different developmental stages.

## Introduction

The World Health Organization (WHO) suggests exclusive breastfeeding as the infant feeding method most appropriate in the first six months of life, the point at which breastfeeding should be combined with complementary feeding up until the age of two or more [1]. These WHO recommendations about breastfeeding are based on its numerous short- and long-term benefits on infants´ health, including improvements in neurophysiological and motor development, cognitive function and intelligence during childhood [2–5]. On the basis of these studies, the duration and type of breastfeeding -exclusive or mixed- necessary to achieve neurophysiological improvements remain to be clarified; however, Wigg et al. [6] showed higher intelligence levels in those children who had been exclusively breastfed at 6 months in contrast with those who had never been breastfed. Additionally, breastfeeding duration has been positively associated with intelligence in young adult life, reporting significantly higher scores on intelligence tests in those young adults who were breastfed for more than 6 months compared to those who were for less than 6 months [7].

Brain-derived neurotrophic factor (BDNF) is a small dimeric protein that is mainly expressed in the hippocampus [8], which promotes synaptic connections and is involved in the growth, development, maintenance and survival of the central nervous system [9, 10]. BDNF has an essential role in dendrite formation and differentiation, and in plasticity [11]. Additionally, BDNF is able to cross the blood-brain barrier, thus allowing both to establish the BDNF levels throughout blood analytic determinations and to relate the BNDF levels in the central nervous system to those in serum [12]. However, the relation between both compartments could be affected by the peripheral and non-cerebral synthesis [13].

In this sense, the content of BDNF in human milk might explain the above-mentioned contribution of breastfeeding to neurological development in the first years of life [14, 15]. In fact, a positive association between breastfeeding and both serum BDNF and neuronal development in infants between four and six months of age has been reported [16]. However, whether this relationship is maintained until school age has not yet been elucidated.

The importance of BDNF lies in its role in learning and memory in childhood, specifically due to its function in long-term memory (LTM) development [17]; yet, the underlying cellular processes remain unknown [18]. At the clinical level, research is needed to clarify if the relationship between BDNF and breastfeeding is maintained over an individual´s lifetime, and therefore, for how long breastfeeding is involved in learning, and LTM development and/or maintenance.

Due to the facts that, first, infancy is a critical period for important development and for the acquisition of cognitive skills [19], second, BDNF acquires an essential function in children cognitive development [17], and third, the breastfeeding effect over other cognitive development-related outcomes is maintained from birth through childhood [20, 21], assessing whether differences in BDNF levels among breastfed and non-breastfed infants persist over childhood is necessary. Thus, this study aimed to assess the relationship between exclusive breastfeeding and BDNF serum levels in Spanish schoolchildren.

## Materials and methods

### Study design

This cross-sectional study is based on the baseline data derived from a cluster randomized controlled trial (registration number NCT03236337), which aimed to assess the effectiveness of an after-school physical activity intervention (MOVI-daFit!) on reducing fat mass and cardiovascular risk, and improving fitness and cognition in children. Two randomly assigned parallel groups were established; on one side, the MOVI-daFit! intervention group, which participated in 60-minute after-school sessions 4 days per week, following a game program based on high-intensity interval training; on the other hand, both intervention and control group received physical education sessions in accordance with Spanish schools´ legal requirements [22]. Data collection was carried out between September 2017 and June 2018. A detailed methodological description is reported elsewhere [22].

### Study sample

The MOVI-daFit! study included 570 children aged eight to 11 years old from 10 schools in Cuenca, a province in Spain. BDNF was measured in a subsample of 220 randomly selected children. Only children with data regarding breastfeeding duration and with BDNF serum levels were included in this study (n = 202). The study protocol was approved by the Clinical Research Ethics Committee of the *Virgen de la Luz hospital*, Cuenca (REG: 2016/PI021). The research team presented the objectives and procedures of the study to the school boards to obtain approval. Parents were asked to sign an informed consent for their children´s participation in the study, who gave their verbal consent when their collaboration was requested.

### Exclusive breastfeeding assessment

At the same time that the rest of the variables were measured in children, data on exclusive breastfeeding were collected from mothers by using a detailed breastfeeding assessment scale completed at home (available as S1 File), which was developed as part of this study, since questionnaires for measuring type and duration of breastfeeding whose validity and rationale had been previously published were not identified. Nonetheless, a pre-test over more than 100 participants from different sociodemographic status was carried out in order to test both readability and clarity of the scale. Likewise, 15 of those participants were required to conduct in-depth interviews to verify if the data from the personal interview corresponded to the previous information provided in the scale.

In this assessment scale, mothers specified the type of feeding their children received each month in the first 24 months of life, and the time during which their child had received breastfeeding, formula or complementary feeding (liquid or solid nutrition other than breast milk). Mothers were able to list multiple options as necessary. (Available as S1 File)

The duration of exclusive breastfeeding was calculated, and categorized into: (i) no exclusive breastfeeding including children who were exclusively formula-fed; (ii) exclusive breastfeeding for ≤6 months; and (iii) exclusive breastfeeding for >6 months. In the third category it is assumed that complementary feeding is introduced at 6 months of age, so those children who were fed with both exclusive breastfeeding and complementary feeding were only included, excluding those who were with formula feeding or mixed breastfeeding and complementary feeding.

## Quantitative determination of BDNF serum levels

BDNF levels were determined within the lipid profile as the primary outcome using a 12-hours fasting blood sample. Blood specimen were collected from the cubital vein between 9:00 and 10:00 AM, with two aliquots being obtained from each participant, so that one sample could be frozen for future analyses that could be of interest to parents. BDNF serum levels were determined using an ELISA method (BDNF ELISA kit SK00752-01, Aviscera Biosciences, Santa Clara, CA, USA), after the appropriate dilution of samples (1:100). All assays were performed in duplicate using the buffers, diluents and substrates provided by the manufacturer.

Briefly, standards and samples were added to pre-coated 96-well flat-bottom plates and shaken for two hours at room temperature. Subsequently, after washing four times, detection antibody was added to each well and plates were incubated for two hours in constant shaking at room temperature. Following another wash step, streptavidin conjugated with horseradish-peroxidase was added, and plates were incubated for 60 minutes at room temperature protected from light. Unbound streptavidin was discarded and TMB substrate solution was added. The reaction was stopped 15 minutes later. Absorbance was read at 450 nm on an iMart microplate reader (BioRad, Hercules, CA, USA) and BDNF concentrations were determined according to the BDNF standard curve (ranging from 23.4 to 750.0 pg/mL).

## Anthropometric assessment

Anthropometric variables were measured twice by trained nurses using standardized procedures with the average being used for the statistical analyses. Weight and height were measured using a scale (Seca 861) and a wall stadiometer (Seca 222), respectively. In both measures, children were required to wear light clothing and to be barefoot.

Body mass index (BMI) was calculated using weight (kg)/height (m)$^2$. Waist circumference was calculated as the average of two measurements at the end of expiration in the mid-point between the iliac crest and the costal margin when the child was upright using a tape measure. Body fat percentage was estimated with an eight-electrode Tanita Segmental-418 bio impedance analysis system (Tanita Corp., Tokyo, Japan) [23].

## Potential confounding factors

The following breastfeeding related variables were considered as potential confounders: age, birth weight (reported by parents), family socioeconomic status (SES) (using the Spanish Epidemiology Society scale [24], which takes into account the parents' educational level and employment status) and children´s sexual maturation (reported by parents using Tanner stages [25, 26] to identify pubertal status).

## Statistical analyses

Interval scale variables were checked for normal distribution through both graphical procedures and using the Kolmogorov-Smirnov test. Data are presented as mean (standard deviation—SD) for continuous variables, and as counts and percentages for categorical variables. Characteristics of participants were compared by sex using the Fisher´s exact test for categorical variables and Student's t test for continuous variables.

Covariance analysis (ANCOVA) was used to test differences in mean BDNF serum levels by exclusive breastfeeding duration categories. Firstly, ANCOVA was stratified by sex and controlled for age, birth weight, SES and sexual maturation. Secondly, the analysis was stratified by age, controlling for sex, birth weight, SES and sexual maturation. The mean differences

in BDNF serum levels according to age categories and sexual maturation stages controlling for sex, age, birth weight and SES were also tested.

All statistical analyses were performed using IBM SPSS 25.0 Statistics software, and the level of significance was set at $\alpha < 0.05$.

## Results

### Characteristics of study participants

This study included 202 children aged between eight and 11 years (mean = 9.60, SD = 0.69), of which 49.5% (n = 100) were boys. Participants´ characteristics were compared by sex (Table 1). Statistically significant differences were found for both body fat percentage and birth weight, with body fat percentage being higher in girls than in boys, while the opposite was observed for birth weight ($p < 0.05$). Regarding the infant feeding method, 43 children (21.3%) were never breastfed, and the remaining 159 children (78.7%) were breastfed; of which, 139 (87.42%) were exclusively breastfed for six months or less. No statistically

**Table 1. Characteristics of the study sample by sex.**

|  | Total (n = 202) | Boys (n = 100) | Girls (n = 102) | p-Value |
|---|---|---|---|---|
| Age (years) | 9.60 (0.69) | 9.61 (0.69) | 9.60 (0.69) | .903 |
| Physical characteristics |  |  |  |  |
| Weight (kg) | 37.31 (9.96) | 37.31 (9.34) | 37.31 (10.59) | .998 |
| Height (cm) | 141.39 (7.63) | 141.52 (7.07) | 141.25 (8.18) | .799 |
| BMI (kg/m²) | 18.47 (3.73) | 18.48 (3.72) | 18.46 (3.75) | .971 |
| BF % | 24.33 (6.46) | 22.99 (6.48) | 25.65 (6.20) | **.003** |
| Waist circumference (cm) | 66.40 (9.48) | 67.16 (9.46) | 65.67 (9.48) | .264 |
| Birth weight (kg) | 3.22 (0.57) | 3.34 (0.52) | 3.10 (0.60) | **.004** |
| Mothers´ gestational age (weeks) | 38.75 (2.43) | 38.95 (2.09) | 38.54 (2.74) | .265 |
| Exclusive breastfeeding |  |  |  |  |
| No exclusive breastfeeding | 43 (21.3) | 20 (20.0) | 23 (22.5) | .590 |
| ≤6 months | 139 (68.8) | 68 (68.0) | 71 (69.6) |  |
| >6 months | 20 (9.9) | 12 (12.0) | 8 (7.8) |  |
| BDNF (nmol/ml) | 47.82 (9.90) | 46.83 (8.99) | 48.79 (10.67) | .159 |
| SES |  |  |  |  |
| Low | 4 (2.2) | 2 (2.2) | 2 (2.2) | .426 |
| Medium-low | 42 (23.0) | 23 (25.3) | 19 (20.7) |  |
| Medium | 98 (53.6) | 52 (57.1) | 46 (50.0) |  |
| Medium-high | 36 (19.7) | 13 (14.3) | 23 (25.0) |  |
| High | 3 (1.6) | 1 (1.1) | 2 (2.2) |  |
| Sexual maturation (Tanner stages) |  |  |  |  |
| Pre-pubertal | 68 (44.7) | 32 (43.2) | 36 (46.2) | .475 |
| Early-pubertal | 54 (35.5) | 30 (40.5) | 24 (30.8) |  |
| Mid-pubertal | 26 (17.1) | 10 (13.5) | 16 (20.5) |  |
| Late-pubertal | 3 (2.0) | 1 (1.4) | 2 (2.6) |  |
| Post-pubertal | 1 (0.7) | 1 (1.4) | 0 (0.0) |  |

Data are exposed by mean ± standard deviation, except for frequency variables (exclusive breastfeeding, SES and sexual maturation) which are shown as n (%). The values in bold indicate a statistical significance for $p < 0.05$, analyzed by Student's t test (continuous variables) or Fisher´s exact test (categorical variables).

Data about participants in SES and sexual maturation variables show missing of 9.4 and 25%, respectively.

BMI, body mass index; BF %, body fat percentage; BDNF, brain-derived neurotrophic factor; SES, socioeconomic status.

**Table 2. Mean difference in BDNF levels by exclusive breastfeeding categories and sex.**

| | n | Exclusive breastfeeding categories | | | p-Value |
| --- | --- | --- | --- | --- | --- |
| | | No exclusive breastfeeding | ≤ 6 months | > 6 months | |
| BDNF levels (nmol/ml) | | | | | |
| Boys | 68 | 46.34 (10.58) | 46.14 (8.46) | 49.41 (7.08) | .871 |
| Girls | 76 | 48.33 (9.88) | 50.16 (10.90) | 46.00 (7.13) | .245 |
| Total | 144 | 47.48 (10.08) | 48.23 (9.96) | 47.86 (6.97) | .943 |

Data are presented by mean (± standard deviation).

BDNF, brain-derived neurotrophic factor

Analysis adjusted for age, birth weight, socioeconomic status and sexual maturation.

significant association was found between sex and exclusive breastfeeding categories (p = 0.590). Finally, no significant differences were observed in BDNF, age, anthropometric characteristics, birth weight, mothers´ gestational age, SES and sexual maturation between children who had information on breastfeeding and those who did not (S1 Table, available as S1 File).

### Exclusive breastfeeding and BDNF serum levels

The mean differences in BDNF serum levels between breastfeeding categories, by sex, are shown in Table 2. No significant differences were found in BDNF (before and after controlling for age, birth weight, SES and sexual maturation) among children depending on the breastfeeding category (not exclusively breastfed, exclusively breastfed for six months or less and exclusively breastfed for more than six months). Also, no sex differences in BDNF serum levels were observed. Likewise, there were no differences in BDNF serum levels between exclusive breastfeeding categories by age groups Table 3.

Finally, because of collinearity between age and sexual maturation was observed (p = 0.008), only the mean differences in BDNF according to sexual maturation stages are showed. No significant trend was observed according to sexual maturation (Tanner stages) Table 4.

## Discussion

To the best of our knowledge, the present study is the first to investigate the relationship between exclusive breastfeeding and BDNF serum levels in children. Our results showed that both presence and maintenance of exclusive breastfeeding were not significantly associated with BDNF levels in eight- to 11-year-old children. There were no differences in BDNF serum

**Table 3. Mean difference in BDNF levels by exclusive breastfeeding categories and age.**

| | n | Exclusive breastfeeding categories | | | Total | p-Value |
| --- | --- | --- | --- | --- | --- | --- |
| | | No exclusive breastfeeding | ≤ 6 months | > 6 months | | |
| Age (years) | | | | | | |
| 8–9 | 57 | 48.14 (12.17) | 47.45 (8.40) | 56.60 (9.10) | 47.92 (9.29) | .518 |
| 10–11 | 87 | 47.13 (9.09) | 48.84 (11.07) | 45.92 (5.20) | 48.09 (10.08) | .453 |
| p-Value | | .872 | .086 | .486 | .261 | |

Data are presented by mean (± standard deviation).

BDNF, brain-derived neurotrophic factor.

Analysis adjusted for sex, birth weight, socioeconomic status and sexual maturation.

**Table 4. Mean difference in BDNF levels by sexual maturation categories.**

|  | Sexual maturation (Tanner stages) | | | | | |
|---|---|---|---|---|---|---|
|  | Pre-pubertal (n = 65) | Early-pubertal (n = 51) | Mid-pubertal (n = 24) | Late-post pubertal (n = 4) | Total (n = 144) | p-Value |
| BDNF levels (nmol/ml) | 48.05 (9.86) | 47.98 (10.73) | 47.20 (7.49) | 53.05 (8.02) | 48.02 (9.74) | .366 |

Data are presented by mean (± standard deviation).

BDNF, brain-derived neurotrophic factor.

Analysis adjusted for sex, age, birth weight and socioeconomic status.

levels between children who were not exclusively breastfed and those who were breastfed for six months or less and for more than six months. Additionally, we did not observe any differences between boys and girls regarding the association among exclusive breastfeeding and BNDF levels, and our findings did not support any trend in BDNF levels according to age or sexual stages maturation trend.

Despite existing evidence of BDNF in human milk from lactating women, which acts as a growth and development factor of the central nervous system and is involved in the persistence of certain primary sensory neurons [27], studies addressing the relationship between breastfeeding and BDNF levels throughout childhood are scarce [14, 28, 29]. Nevertheless, numerous studies have reported the influence of BDNF levels in different health-based outcomes, such as obesity [30], autism [31], cognitive function [16, 32, 33] and attention deficit-hyperactivity disorder [34, 35]. Likewise, exogenous intake of BDNF in rodents by neural injection has been analyzed, and visible improvements in cognitive achievement and a decline when BDNF is blocked in the hippocampus have been observed [36].

Our findings differ from the only previous research analyzing the relationship between breastfeeding and BDNF levels [16]. Using a sample of infants aged between four and six months, Nassar et al. [16] reported that the group who was exclusively breastfed had significantly higher BDNF levels compared to the formula fed and the mixed-fed groups. Thus, it may be that the association between breastfeeding and BDNF levels is only observed at an early age. Thereby, the influence of breastfeeding on BDNF levels is diluted throughout life, potentially due to the influence of environmental and behavioral factors [37]. More research is required to identify the determinants that affect BDNF levels and in what proportion from infancy to adolescence.

Intestinal permeability in neonates, which would positively influence the increase of BDNF levels in this age group, may explain why the effect of breastfeeding is not maintained from birth to adolescence. This effect is likely more plausible in the first months of life, coinciding with the period of maximum brain development [15]. Additionally, both the range of BDNF plasma levels and variability in the BDNF measurement may explain the lack of significant associations observed in this study [38, 39]. Moreover, although BDNF protein is mainly expressed in the hippocampus and then it crosses the blood-brain barrier [8, 12], it is also expressed in non-cerebral tissues [13]. As such, the correlation between the levels of the two compartments may not be very strong and, therefore, the BDNF levels measured in our study may not represent faithfully its proportion in the cerebral area. Finally, platelet count in children could also influence BDNF levels since most BDNF is stored in platelets [40].

The influence of hormonal status on BDNF levels among individuals could also explain the results obtained in our study. BDNF is negatively associated with age, being higher in children than in adolescents [41, 42]. Therefore, the association between breastfeeding and BDNF may be diluted according to hormonal development, with a more significant association at early ages [16]. Likewise, our results may be affected by the fact that BDNF fluctuates depending on

certain pathologies, caloric restriction or exercise, which cause a similar effect to that of BDNF in human milk [37]. Finally, recall bias regarding breastfeeding could influence our results since we included children aged between eight and 11 years, and the reliability of maternal recall has been shown to be until six years after childbirth [43].

Our study has some limitations, which must be considered: (i) inherent limitations in its design, since the retrospective and cross-sectional design does not allow causal relationships to be established and consequently, the findings could be interfered; (ii) although our total sample was higher than that of previous studies, some subgroups did not have a sufficient sample size to reach statistical significance; for instance, regarding exclusive breastfeeding categorization, and with the purpose of working with representative sample sizes, the 'exclusive breastfeeding ≤ 6 months' category included a great range of breastfeeding periods (from one to six months), not being able to compare the optimal exclusive breastfeeding duration (= 6 months, according WHO recommendations) with the rest of periods; (iii) a variety of validated methods for measuring serum BDNF content are currently available [38], and BDNF concentration could depend to some extent on the laboratory method used to measure it; (iv) maternal responses about the type and duration of breastfeeding could be affected by recall bias; and (v) the development and implementation of breastfeeding questionnaires entails an inherent complexity of the infant feeding process itself. The limitations must be considered to interpret cautiously the findings of this study, and further research which could solve the limitations of this study is necessary. Longitudinal designs and rigorous registers in clinical records about breastfeeding type and duration from birth could facilitate the attainment of reliable findings.

## Conclusions

In conclusion, our study does not support that the effect of breastfeeding on BDNF levels persist until pre-pubertal age. However, our results should be confirmed by future studies using reliable methods to measure both breastfeeding exposure and BDNF serum levels, as well as to determine the influence of children´s growth and hormonal development on BDNF.

## Supporting information

**S1 Data.**
(XLSX)

**S1 File.**
(DOCX)

## Acknowledgments

We would like to thank to all schools, families and pupils for their enthusiastic participation in the study.

## Author Contributions

**Conceptualization:** Carlos Berlanga-Macías, Mairena Sánchez-López, Montserrat Solera-Martínez, Ana Díez-Fernández, Inmaculada Ballesteros-Yáñez, Carlos A. Castillo-Sarmiento, Isabel A. Martínez-Ortega, Vicente Martínez-Vizcaíno.

**Data curation:** Carlos Berlanga-Macías, Mairena Sánchez-López, Montserrat Solera-Martínez, Ana Díez-Fernández, Inmaculada Ballesteros-Yáñez, Carlos A. Castillo-Sarmiento, Isabel A. Martínez-Ortega, Vicente Martínez-Vizcaíno.

**Formal analysis:** Carlos Berlanga-Macías, Mairena Sánchez-López, Montserrat Solera-Martínez, Ana Díez-Fernández, Inmaculada Ballesteros-Yáñez, Carlos A. Castillo-Sarmiento, Isabel A. Martínez-Ortega, Vicente Martínez-Vizcaíno.

**Funding acquisition:** Carlos Berlanga-Macías, Mairena Sánchez-López, Montserrat Solera-Martínez, Ana Díez-Fernández, Inmaculada Ballesteros-Yáñez, Carlos A. Castillo-Sarmiento, Isabel A. Martínez-Ortega, Vicente Martínez-Vizcaíno.

**Investigation:** Carlos Berlanga-Macías, Mairena Sánchez-López, Montserrat Solera-Martínez, Ana Díez-Fernández, Inmaculada Ballesteros-Yáñez, Carlos A. Castillo-Sarmiento, Isabel A. Martínez-Ortega, Vicente Martínez-Vizcaíno.

**Methodology:** Carlos Berlanga-Macías, Mairena Sánchez-López, Montserrat Solera-Martínez, Ana Díez-Fernández, Inmaculada Ballesteros-Yáñez, Carlos A. Castillo-Sarmiento, Isabel A. Martínez-Ortega, Vicente Martínez-Vizcaíno.

**Project administration:** Carlos Berlanga-Macías, Mairena Sánchez-López, Montserrat Solera-Martínez, Ana Díez-Fernández, Inmaculada Ballesteros-Yáñez, Carlos A. Castillo-Sarmiento, Isabel A. Martínez-Ortega, Vicente Martínez-Vizcaíno.

**Resources:** Carlos Berlanga-Macías, Mairena Sánchez-López, Montserrat Solera-Martínez, Ana Díez-Fernández, Inmaculada Ballesteros-Yáñez, Carlos A. Castillo-Sarmiento, Isabel A. Martínez-Ortega, Vicente Martínez-Vizcaíno.

**Software:** Carlos Berlanga-Macías, Mairena Sánchez-López, Montserrat Solera-Martínez, Ana Díez-Fernández, Inmaculada Ballesteros-Yáñez, Carlos A. Castillo-Sarmiento, Isabel A. Martínez-Ortega, Vicente Martínez-Vizcaíno.

**Supervision:** Carlos Berlanga-Macías, Mairena Sánchez-López, Montserrat Solera-Martínez, Ana Díez-Fernández, Inmaculada Ballesteros-Yáñez, Carlos A. Castillo-Sarmiento, Isabel A. Martínez-Ortega, Vicente Martínez-Vizcaíno.

**Validation:** Carlos Berlanga-Macías, Mairena Sánchez-López, Montserrat Solera-Martínez, Ana Díez-Fernández, Inmaculada Ballesteros-Yáñez, Carlos A. Castillo-Sarmiento, Isabel A. Martínez-Ortega, Vicente Martínez-Vizcaíno.

**Visualization:** Carlos Berlanga-Macías, Mairena Sánchez-López, Montserrat Solera-Martínez, Ana Díez-Fernández, Inmaculada Ballesteros-Yáñez, Carlos A. Castillo-Sarmiento, Isabel A. Martínez-Ortega, Vicente Martínez-Vizcaíno.

**Writing – original draft:** Carlos Berlanga-Macías, Mairena Sánchez-López, Montserrat Solera-Martínez, Ana Díez-Fernández, Inmaculada Ballesteros-Yáñez, Carlos A. Castillo-Sarmiento, Isabel A. Martínez-Ortega, Vicente Martínez-Vizcaíno.

**Writing – review & editing:** Carlos Berlanga-Macías, Mairena Sánchez-López, Montserrat Solera-Martínez, Ana Díez-Fernández, Inmaculada Ballesteros-Yáñez, Carlos A. Castillo-Sarmiento, Isabel A. Martínez-Ortega, Vicente Martínez-Vizcaíno.

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
