## [Decision Letter · Decision Letter 0]

30 Dec 2020

PONE-D-20-34834

Relationship between exclusive breastfeeding and brain-derived neurotrophic factor in children

PLOS ONE

Dear Dr. Solera-Martínez,

Thank you for submitting your manuscript to PLOS ONE. After careful consideration, we feel that it has merit but does not fully meet PLOS ONE’s publication criteria as it currently stands. Therefore, we invite you to submit a revised version of the manuscript that addresses the points raised during the review process.

The manuscript  is a good piece of work, with original results. Some important improvements have been suggested by the reviewers.

We look forward to receiving your revised manuscript.

Kind regards,

Marly A. Cardoso, Ph.D.

Academic Editor

PLOS ONE

Journal Requirements:

3. Please include additional information regarding the survey or questionnaire used in the study and ensure that you have provided sufficient details that others could replicate the analyses. For instance, if you developed a questionnaire as part of this study and it is not under a copyright more restrictive than CC-BY, please include a copy, in both the original language and English, as Supporting Information, or include a citation if it has been published previously.

4. In the Methods, please discuss whether and how the questionnaire was validated and/or pre-tested. If these did not occur, please provide the rationale for not doing so.

Reviewers' comments:

Reviewer's Responses to Questions

**Comments to the Author**

1. Is the manuscript technically sound, and do the data support the conclusions?

Reviewer #1: Partly

Reviewer #2: Yes

Reviewer #3: Partly

2. Has the statistical analysis been performed appropriately and rigorously? 

Reviewer #1: Yes

Reviewer #2: Yes

Reviewer #3: No

3. Have the authors made all data underlying the findings in their manuscript fully available?

Reviewer #1: Yes

Reviewer #2: Yes

Reviewer #3: Yes

4. Is the manuscript presented in an intelligible fashion and written in standard English?

Reviewer #1: Yes

Reviewer #2: Yes

Reviewer #3: Yes

5. Review Comments to the Author

Reviewer #1: This paper investigated if exclusive breastfed children in the beginning of life have different serum BDNF levels at the age of 8-11 years old. The participants were children from a Spanish randomized controlled trial. The data was collected in different moments by using trained team, assessment scales (Tanner and breastfeeding history) and blood samples. The findings suggest that the effects of exclusive breastfeeding in the beginning of life on serum BDNF levels do not persist at the age of 8 to 11 years old. Even so, future research should examine this influence using more reliable methods.

The study has some interesting findings and may contribute for the children's heath and development research area. The manuscript is well written, structured and organized, leading to no ambiguity, doubts or demanding extra attention to interpret the information. However, some minor issues should be reviewed.

The introduction section brings a great set of information about the topic and provides a contextualization of the subject to the reader. However, it doesn’t provide a solid base for the hypothesis of the study. Some examples may illustrate this issue:

1. The authors explore the importance of breastfeeding, but it doesn’t mention why the optimal duration of exclusive breastfeeding (until 6 months) would be relevant to BDNF levels or to the neurophysiological development.

2. Also, despite the fact that the study of Poduslo & Curran (reference 12) is very robust, the sample was constituted by adult rats, so using this reference to affirm that serum BDNF is correlated to BDNF levels in central nervous system in the present study population (humans from 8 to 11 years old) is questionable. In addition to that, in the discussion section the authors state as a limitation of the study (lines 262 to 266) that BDNF levels measured may not represent faithfully its proportion in the cerebral area since it is expressed in the hippocampus and in non-cerebral tissues. Therefore, the objective of measuring serum BDNF in the introduction is not clear.

The methodology section evidences the power of this study, providing a good perspective of the study design to the reader. Both statistical analysis and the process to determine BDNF serum levels are detailed and well described in the manuscript, contributing to the credibility of the results. Nevertheless, some aspects should be reviewed.

3. The BDNF sample was taken when the children were 8 – 11 years old, but it is not clear when data the breastfeeding assessment scale was completed by the mothers. The authors state that they only included in the study children with breastfeeding data and blood sample, giving the impression that it was collected in different time points. So when were they collected?

4. The categorization of the exclusive breastfeeding duration is a bit confusing: the 2nd group (exclusive breastfeeding for ≤6 months) includes children exclusive breastfed for 1, 2, 3, 4, 5 and 6 months. This is an important limitation because this group include a great range of breastfeeding periods, not distinguishing the optimal duration of it (=6 months). Also, the 3rd group (exclusive breastfeeding for >6 months) has a relevant percentage of children (almost 10%), what is a really vague information and may be alarming since that in this age the complementary feeding should be introduced.

5. It is not clear if the evaluation of the pubertal status according to the Tanner stages was self-reported or physician-observed.

6. The study was a retrospective and cross-sectional study, using a convenience sample. It is important to state this fact as a limitation because it may interfere in the results even after adjustments were made in the statistical analysis. At the end of this review I left an important reference that could be interesting for the study since it is prospective and population based sample evidencing the importance exclusive breastfeeding and anthropometric variables on head circumference (Giacomini et al., 2020).

The results from the study are well described in the text and are presented in tables in a very fine way. The discussion of the findings highlights important previous studies, intertwining the existing literature and the findings from the presented study. Yet, considerations must be taken in account to improve this section.

7. In lines 230, 231 and 232 the authors state that “exclusive breastfeeding duration was not significantly associated with BDNF levels (…)”. However the study isn’t investigating the duration of exclusive breastfeeding (e.g. in the 2nd category the authors grouped all children that were exclusive breastfed for 1, 2, 3, 4, 5 and 6 months old).

8. In lines 234-236 the authors state that “we did not observe any differences between boys and girls, and our findings did not support an age or sexual stages maturation trend”, but doesn’t explain regarding to what there were no differences.

The conclusions were drawn appropriately based on the data presented, which are fully available in the manuscript, being presented during the text, but also in tables. In addition, the decision to present the results that did not correspond to the initial hypothesis demonstrate the integrity of the authors and transparency of the study. However, the limitations (some of it even state by the authors) should be consider.

Giacomini I, Mazzucchetti L, B Lima TA, B Malta M, H Lourenço B, A Cardoso M; MINA-Brazil Study Group. Breastfeeding practices and weight gain predicted head circumference in young Amazonian children. Acta Paediatr. 2020 Aug 4. doi: 10.1111/apa.15517. Epub ahead of print. PMID: 32749721.

Reviewer #2: This study aimed to verify the relationship between exclusive breastfeeding and brain-derived neurotrophic fator (BDNF) serum levels in children aged 8-11 years. The theme is relevant, interesting and, according to the authors, it is the first to investigate the relationship between exclusive breastfeeding and BDNF serum levels in children.

The paper is a good piece of work, well structured, clear and easy to understand. The methods are well described and the results are clearly presented. The discussion of this paper is well structured, considering several and appropriate previous studies. Most references are current and used adequately.

Despite being a very good paper, some considerations must be made: in my opinion, two limitations of the study are important and may weaken the results: 1) the collection of data on the duration of breastfeeding when the child is 8 years old or more certainly results in a recall bias; 2) only 20 children were exclusively breasted for more than 6 months, perhaps not enough to reach statistical significance. I was wondering if reducing the cutoff point to 4 months (instead of 6 months) could not give a different result...

Forthermore, since, according to the authors, the only previous research analyzing the relationship between breastfeeding and BDNF levels was carried out with infants aged four to six months (who may have recently received a significant amount of BDNF from breast milk), what is the biological plausibility of assessing this association in older children (8-11 years old)? It may be important to include this justification/answer in the introduction section.

There are minor points that should be clarified/edited:

- Line 40: “who were exclusively breasfed?”

- Line 178: I suggest writing “Table 1” in parentheses.

- Line 187: I suggest including “(data not shown on Tables)” at the end of sentence.

Reviewer #3: The authors present results of a study of BDNF in about 200 8-11 year old children in Spain and whether or not levels at this age differed between those that were exclusively breastfed and those that were not. In their analysis, they find no significant differences in BDNF levels by breastfeeding subgroups, by sex or by age and also find no significant differences in levels by sexual maturation. The manuscript will be strengthened if the authors consider the following points.

1. Line 105: authors refer the reader to another manuscript to find a detailed description of the study. Authors should also provide a brief description in this manuscript, so readers can get the general idea of the main study without having to go elsewhere to find that information; readers can then go to the other manuscript of they want more than the brief description.

2. Authors should include a supplemental table that shows how individuals in the BDNF subsample (n=202) compared to those that were not selected to provide samples for BDNF measurement.

3. Who determined Tanner stages (since this was used as a covariate in the model)?

4. lines 166-168: This sentence should be rephrased, as it is not clear if stratified analyses were conducted or if interaction terms were utilized.

5. lines 186-187: A supplemental table should be provided to support the statement about lack of significant differences between those with breastfeeding information and those without.

6. Table 1: Sexual maturation appears to be missing for about 25% of the participants. SES also has missing data (though closer to 10%). Authors should at least make note of that in the notes under the table, especially since these variables are used as covariates in later analyses.

7. Table 1: authors use a chi-square test for comparing categorical variables by sex. Some of the cell counts are quite small, which suggests that Fisher's exact test would be more appropriate (overall interpretation does not change, but it is the more appropriate test).

8. Tables 2, 3, and 4: since these results are adjusted for variables that have missing data, authors should make note of how many individuals were actually included in the analysis. Also, as stated in point 4 above, it is not clear if the different analyses presented in Tables 2 and 3 are stratified by sex or age, or if there is an interaction term between breastfeeding categories and sex or age. This is relevant, since the bulk of the participants fall into the <=6 months of exclusive breastfeeding, so there are quite small cell counts in the other two categories. Also, was there any collinearity between sexual maturation and age?

9. Line 288: this concluding sentence should be rephrased, since not being able to reject the null hypothesis does not mean that the null hypothesis is true.

Minor points:

1. line 265: "very closed" - should "closed" be a different word?

2. line 282: I believe "BDN" should be "BDNF"

6. PLOS authors have the option to publish the peer review history of their article (what does this mean?). If published, this will include your full peer review and any attached files.

Reviewer #1: **Yes: **Isabel Giacomini Marques

Reviewer #2: No

Reviewer #3: No

---

## [Author Response · Author response to Decision Letter 0]

2 Feb 2021

Point-by-point response to reviewers’ comments (manuscript number: PONE-D-20-34834)

JOURNAL REQUIREMENTS:

1- Comment: Please include additional information regarding the survey or questionnaire used in the study and ensure that you have provided sufficient details that others could replicate the analyses. For instance, if you developed a questionnaire as part of this study and it is not under a copyright more restrictive than CC-BY, please include a copy, in both the original language and English, as Supporting Information, or include a citation if it has been published previously.

Authors: As required, we have modified the ‘Material and methods’ section, including the following details about the questionnaire used in the study:

Page 6, line 135: “At the same time that the rest of the variables were measured in children, data on exclusive breastfeeding were collected from mothers by using a detailed breastfeeding assessment scale completed at home (available as Supporting Information), which was developed as part of this study, since questionnaires for measuring type and duration of breastfeeding whose validity and rationale had been previously published were not identified. Nonetheless, a pre-test over more than 100 participants from different sociodemographic status was carried out in order to test both readability and clarity of the scale. Likewise, 15 of those participants were required to conduct in-depth interviews to verify if the data from the personal interview corresponded to the previous information provided in the scale.”

2- Comment: In the Methods, please discuss whether and how the questionnaire was validated and/or pre-tested. If these did not occur, please provide the rationale for not doing so.

Authors: Thank you for your recommendation. As suggested, we have modified the ‘Material and methods’ section as it can be appreciated in point 1 above.

REVIEWER 1:

General comments: 

Introduction section

1- Comment: The authors explore the importance of breastfeeding, but it doesn’t mention why the optimal duration of exclusive breastfeeding (until 6 months) would be relevant to BDNF levels or to the neurophysiological development. 

Authors: Thank you for the comments. As recommended, we have modified the ‘Introduction’ section, including the following details about the relationship between optimal duration of exclusive breastfeeding and neurophysiological development: 

Page 4, line 73: “On the basis of these studies, the duration and type of breastfeeding -exclusive or mixed- necessary to achieve neurophysiological improvements remain to be clarified; however, Wigg et al. (6) showed higher intelligence levels in those children who had been exclusively breastfed at 6 months in contrast with those who had never been breastfed. Additionally, breastfeeding duration has been positively associated with intelligence in young adult life, reporting significantly higher scores on intelligence tests in those young adults who were breastfed for more than 6 months compared to those who were for less than 6 months (7)”

Page 5, line 90: “In this sense, the content of BDNF in human milk might explain the above-mentioned contribution of breastfeeding to neurological development in the first years of life (14,15). In fact, a positive association between breastfeeding and both serum BDNF and neuronal development in infants between four and six months of age has been reported (16). However, whether this relationship is maintained until school age has not yet been elucidated.”

2- Comment: Also, despite the fact that the study of Poduslo & Curran (reference 12) is very robust, the sample was constituted by adult rats, so using this reference to affirm that serum BDNF is correlated to BDNF levels in central nervous system in the present study population (humans from 8 to 11 years old) is questionable. In addition to that, in the discussion section the authors state as a limitation of the study (lines 262 to 266) that BDNF levels measured may not represent faithfully its proportion in the cerebral area since it is expressed in the hippocampus and in non-cerebral tissues. Therefore, the objective of measuring serum BDNF in the introduction is not clear.

Authors: Thank you for the interesting comment. As required, we have clarified the objective of measuring serum BDNF levels:

Page 4, line 85: “Additionally, BDNF is able to cross the blood-brain barrier, thus allowing both to establish the BDNF levels throughout blood analytic determinations and to relate the BNDF levels in the central nervous system to those in serum (12). However, the relation between both compartments could be affected by the peripheral and non-cerebral synthesis (13).”

Methodology section

3- Comment: The BDNF sample was taken when the children were 8 – 11 years old, but it is not clear when data the breastfeeding assessment scale was completed by the mothers. The authors state that they only included in the study children with breastfeeding data and blood sample, giving the impression that it was collected in different time points. So when were they collected?

Authors: Thank you for the reviewer’s comment. We have modified methods sections as follows:

Page 6, line 135: “At the same time that the rest of the variables were measured in children, data on exclusive breastfeeding were collected from mothers by using a detailed breastfeeding assessment scale completed at home (available as Supporting Information)”.

4- Comment: The categorization of the exclusive breastfeeding duration is a bit confusing: the 2nd group (exclusive breastfeeding for ≤6 months) includes children exclusive breastfed for 1, 2, 3, 4, 5 and 6 months. This is an important limitation because this group include a great range of breastfeeding periods, not distinguishing the optimal duration of it (=6 months). Also, the 3rd group (exclusive breastfeeding for >6 months) has a relevant percentage of children (almost 10%), what is a really vague information and may be alarming since that in this age the complementary feeding should be introduced.

Authors: Thank you for your useful comments. As recommended, we have modified both methods and limitations sections as follows:

Page 7, line 153: “In the third category it is assumed that complementary feeding is introduced at 6 months of age, so those children who were fed with both exclusive breastfeeding and complementary feeding were only included, excluding those who were with formula feeding or mixed breastfeeding and complementary feeding”.

This categorization was carried out in line with previous studies which analyzed exclusive breastfeeding and with the purpose of keeping an homogeneous language.

Page 15, line 316: “(ii) although our total sample was higher than that of previous studies, some subgroups did not have a sufficient sample size to reach statistical significance; for instance, regarding exclusive breastfeeding categorization, and with the purpose of working with representative sample sizes, the ‘exclusive breastfeeding ≤ 6 months’ category included a great range of breastfeeding periods (from one to six months), not being able to compare the optimal exclusive breastfeeding duration (= 6 months, according WHO recommendations) with the rest of periods;”

5- Comment: It is not clear if the evaluation of the pubertal status according to the Tanner stages was self-reported or physician-observed.

Authors: Thank you; we have stated how the pubertal status was evaluated.

Page 9, line 190: “children´s sexual maturation (reported by parents using Tanner stages (25,26) to identify pubertal status).”

6- Comment: The study was a retrospective and cross-sectional study, using a convenience sample. It is important to state this fact as a limitation because it may interfere in the results even after adjustments were made in the statistical analysis. At the end of this review I left an important reference that could be interesting for the study since it is prospective and population based sample evidencing the importance exclusive breastfeeding and anthropometric variables on head circumference (Giacomini et al., 2020).

Authors: Thank you for both the comment and the reference provided. As suggested, we have modified the limitations sections as follows: 

Page 15, line 314: “(i) inherent limitations in its design, since the retrospective and cross-sectional design does not allow causal relationships to be established and consequently, the findings could be interfered.

Page 16, line 330: “Longitudinal designs and rigorous registers in clinical records about breastfeeding type and duration from birth could facilitate the attainment of reliable findings.”

Results section

7- Comment: In lines 230, 231 and 232 the authors state that “exclusive breastfeeding duration was not significantly associated with BDNF levels (…)”. However the study isn’t investigating the duration of exclusive breastfeeding (e.g. in the 2nd category the authors grouped all children that were exclusive breastfed for 1, 2, 3, 4, 5 and 6 months old).

Authors: Thank you for the interesting comments. As suggested, we have rephrased the sentence about reported findings. 

Page 13, line 264: “Our results showed that both presence and maintenance of exclusive breastfeeding were not significantly associated with BDNF levels in eight- to 11-year-old children.”

8- Comment: In lines 234-236 the authors state that “we did not observe any differences between boys and girls, and our findings did not support an age or sexual stages maturation trend”, but doesn’t explain regarding to what there were no differences.

Authors: We would like to thank the thoughtful comment. As recommended, a more complete description about findings has been included.

Page 13, line 269: “Additionally, we did not observe any differences between boys and girls regarding the association among exclusive breastfeeding and BNDF levels, and our findings did not support any trend in BDNF levels according to age or sexual stages maturation trend.”

Discussion section

9- Comment: The conclusions were drawn appropriately based on the data presented, which are fully available in the manuscript, being presented during the text, but also in tables. In addition, the decision to present the results that did not correspond to the initial hypothesis demonstrate the integrity of the authors and transparency of the study. However, the limitations (some of it even state by the authors) should be consider.

Authors: As suggested, limitations section has been improved.

Page 16, line 328: “The limitations must be considered to interpret cautiously the findings of this study, and further research which could solve the limitations of this study is necessary.”

REVIEWER 2:

General comments: 

1- Comment: The collection of data on the duration of breastfeeding when the child is 8 years old or more certainly results in a recall bias.

Authors: Thanks for this comment. We agree with your assessment, so this limitation has been stated on limitation section and it must be considered in order to carry out further research. 

Page 16, line 325: “(iv) maternal responses about the type and duration of breastfeeding could be affected by recall bias;”

Page 16, line 328: “The limitations must be considered to interpret cautiously the findings of this study, and further research which could solve the limitations of this study is necessary. Longitudinal designs and rigorous registers in clinical records about breastfeeding type and duration from birth could facilitate the attainment of reliable findings.”

2- Comment: Only 20 children were exclusively breasted for more than 6 months, perhaps not enough to reach statistical significance. I was wondering if reducing the cutoff point to 4 months (instead of 6 months) could not give a different result....

Authors: Thank you for the interesting comment. It is true that sample size of some subgroups could be not enough to reach statistical significance, among which is the ‘exclusive breastfeeding for more than 6 months’ group. However, in accordance with WHO recommendations (1), the cut-off point at 6 months has been stablished. Only then could we both determine the total account of population who adheres to WHO recommendations and assess if complying with these recommendations had a positive and significant effect on BDNF levels.

On the other hand, relevant studies which have evaluated exclusive breastfeeding have established the cut-off point at 6 months (2, 3, 4), as it is the case of a recent study whose aim was to test the validity and reliability of a breastfeeding questionnaire (5). Following this trend set by these studies, among others, to stablish the cut-off point at 6 months could be appropriated.

1- World Health Organization. Infant and young child feeding. Geneva, Switzerland: World Health Organization; 2017.

2- Victora CG, Bahl R, Barros AJD, França GVA, Horton S, Krasevec J, et al. Breastfeeding in the 21st century: Epidemiology, mechanisms, and lifelong effect. Lancet. 2016;387(10017):475–490. DOI: 10.1016/S0140-6736(15)01024-7

3- Bhattacharjee NV, Schaeffer LE, Marczak LB, Ross JM, Swartz SJ, Albright J, et al. Mapping exclusive breastfeeding in Africa between 2000 and 2017. Nature Medicine. 2019; 25(8):1205-1212. DOI: 10.1038/s41591-019-0525-0

4- Wigg NR, Tong S, McMichael AJ, Baghurst PA, Vimpani G, Roberts R. Does breastfeeding at six months predict cognitive development? Aust N Z J Public Health. 1998;22(2):232-6. DOI: 10.1111/j.1467-842x.1998.tb01179.x

5- Davie P, Bick D and Chilcot J. The Beliefs About Breastfeeding Questionnaire (BAB‐Q): A psychometric validation study. Br J Health Psychol. 2020. DOI: 10.1111/bjhp.12498 

3- Comment: Furthermore, since, according to the authors, the only previous research analyzing the relationship between breastfeeding and BDNF levels was carried out with infants aged four to six months (who may have recently received a significant amount of BDNF from breast milk), what is the biological plausibility of assessing this association in older children (8-11 years old)? It may be important to include this justification/answer in the introduction section.

Authors: We would like to thank the thoughtful comment. As recommended, a more complete justification in the introduction section has been included.

Page 5, line 102: “Due to the facts that, first, infancy is a critical period for important development and for the acquisition of cognitive skills (19), second, BDNF acquires an essential function in children cognitive development (17), and third, the breastfeeding effect over other cognitive development-related outcomes is maintained from birth through childhood (20,21), assessing whether differences in BDNF levels among breastfed and non-breastfed infants persist over childhood is necessary.”

Minor comments: 

4- Comment: Line 40: “who were exclusively breastfed?”

Authors: Done.

Page 2, line 39: “ANCOVA models showed no significant differences in BDNF levels between children who were exclusively breastfed for more than six months versus those who were not (p > 0.05).”

5- Comment: Line 178: I suggest writing “Table 1” in parentheses.

Authors: Thank you. As suggested, we have modified the text as follows:

Page 10, line 209: “This study included 202 children aged between eight and 11 years (mean = 9.60, SD = 0.69), of which 49.5% (n = 100) were boys. Participants´ characteristics were compared by sex (Table 1).”

6- Comment: Line 187: I suggest including “(data not shown on Tables)” at the end of sentence.

Authors: Thank you for the comment. A supplemental table to support the statement about lack of significant differences between those who had breastfeeding information and those who did not has been included in the manuscript, and the text has been modified as follows:

Page 10, line 217: “Finally, no significant differences were observed in BDNF, age, anthropometric characteristics, birth weight, mothers´ gestational age, SES and sexual maturation between children who had information on breastfeeding and those who did not (Table S1, available as Supporting Information).”

REVIEWER 3:

General comments:

1- Comment: Line 105: authors refer the reader to another manuscript to find a detailed description of the study. Authors should also provide a brief description in this manuscript, so readers can get the general idea of the main study without having to go elsewhere to find that information; readers can then go to the other manuscript of they want more than the brief description.

Authors: Thank you for the comment. As suggested, we have included a brief description of the above-mentioned manuscript as follows:

Page 6, line 116: “Two randomly assigned parallel groups were established; on one side, the MOVI-daFit! intervention group, which participated in 60-minute after-school sessions 4 days per week, following a game program based on high-intensity interval training; on the other hand, both intervention and control group received physical education sessions in accordance with Spanish schools’ legal requirements (22).”

2- Comment: Authors should include a supplemental table that shows how individuals in the BDNF subsample (n=202) compared to those that were not selected to provide samples for BDNF measurement.

Authors: A representative subsample of 220 children, in which BDNF was measured, was randomly selected out of 570 children who participated in MOVI-daFit! study. Therefore, it is taken for granted that there were not statistically significant differences among children who were selected and those who were not.

3- Comment: Who determined Tanner stages (since this was used as a covariate in the model)?

Authors: Thank you for your comment. As recommended, we have included who determined Tanner stages as follows: 

Page 9, line 190: “[…] children´s sexual maturation (reported by parents using Tanner stages (25,26) to identify pubertal status).”

4- Comment: Lines 166-168: This sentence should be rephrased, as it is not clear if stratified analyses were conducted or if interaction terms were utilized.

Authors: We would like to thank the thoughtful comment. As suggested, a more complete description of statistical analysis section has been included.

Page 9, line 198: “Covariance analysis (ANCOVA) was used to test differences in mean BDNF serum levels by exclusive breastfeeding duration categories. Firstly, ANCOVA was stratified by sex and controlled for age, birth weight, SES and sexual maturation. Secondly, the analysis was stratified by age, controlling for sex, birth weight, SES and sexual maturation.”

5- Comment: Lines 186-187: A supplemental table should be provided to support the statement about lack of significant differences between those with breastfeeding information and those without.

Authors: Thank you for the comment. As recommended, a supplemental table to support the statement about lack of significant differences between those who had breastfeeding information and those who did not has been included in the manuscript, and the text has been modified as follows:

Page 10, line 217: “Finally, no significant differences were observed in BDNF, age, anthropometric characteristics, birth weight, mothers´ gestational age, SES and sexual maturation between children who had information on breastfeeding and those who did not (Table S1, available as Supporting Information).” 

6- Comment: Table 1: Sexual maturation appears to be missing for about 25% of the participants. SES also has missing data (though closer to 10%). Authors should at least make note of that in the notes under the table, especially since these variables are used as covariates in later analyses.

Authors: Thank you for the thoughtful comment. As suggested, we have modified the notes under the Table 1.

Page 11, line 225: “Data about participants in SES and sexual maturation variables show missing of 9.4 and 25%, respectively.”

7- Comment: Table 1: authors use a chi-square test for comparing categorical variables by sex. Some of the cell counts are quite small, which suggests that Fisher's exact test would be more appropriate (overall interpretation does not change, but it is the more appropriate test).

Authors: After categorical variables were compared by sex using both Fisher´s exact test and chi-square test, no differences among obtained data were found. As recommended, Fisher´s exact test was used:

Page 9, line 196: Characteristics of participants were compared by sex using the Fisher´s exact test for categorical variables and Student’s t test for continuous variables.

8- Comment: Tables 2, 3, and 4: since these results are adjusted for variables that have missing data, authors should make note of how many individuals were actually included in the analysis. 

Authors: Thank you for the recommendation. As suggested, Tables 2,3,4 have been modified and an update of how many individuals were included has been noted (Pages 12 and 13: Tables 2,3, 4).

9- Comment: Also, as stated in point 4 above, it is not clear if the different analyses presented in Tables 2 and 3 are stratified by sex or age, or if there is an interaction term between breastfeeding categories and sex or age. This is relevant, since the bulk of the participants fall into the <=6 months of exclusive breastfeeding, so there are quite small cell counts in the other two categories.

Authors: See point 4 above.

10- Comment: Also, was there any collinearity between sexual maturation and age?

Authors: Thank you for the comment. After correlation was carried out, collinearity between sexual maturation and age was identified (Spearman´s correlation coefficient = 0.21; p=0.008). For this reason, we only showed the mean differences in BDNF levels according to sexual maturation. 

Page 12, line 235: “Finally, because of collinearity between age and sexual maturation was observed (p=0.008), only the mean differences in BDNF according to sexual maturation stages are showed. No significant trend was observed according to sexual maturation (Tanner stages) Table 4.”

11- Comment: Line 288: this concluding sentence should be rephrased, since not being able to reject the null hypothesis does not mean that the null hypothesis is true.

Authors: Thank you for your comment. As suggested, we have rephrased the above-mentioned sentence as follows:

Page 16, line 334: “In conclusion, our study does not support that the effect of breastfeeding on BDNF levels persist until pre-pubertal age.”

Minor comments: 

12- Comment: Line 265: "very closed" - should "closed" be a different word?

Authors: As recommended, another different word has been included. 

Page 15, line 300: “As such, the correlation between the levels of the two compartments may not be very strong and, therefore, the BDNF levels measured in our study may not represent faithfully its proportion in the cerebral area.”

13- Comment: Line 282: I believe "BDN" should be "BDNF".

Authors: Thank you for your appreciation. It was a mistake, but it has been already rectified.

Page 16, line 323: “(iii) a variety of validated methods for measuring serum BDNF content are currently available (38)”.

We sincerely appreciate the fruitful comments of the reviewers.

---

## [Decision Letter · Decision Letter 1]

18 Feb 2021

Relationship between exclusive breastfeeding and brain-derived neurotrophic factor in children

PONE-D-20-34834R1

Dear Dr. Solera-Martínez,

We’re pleased to inform you that your manuscript has been judged scientifically suitable for publication and will be formally accepted for publication once it meets all outstanding technical requirements.

Kind regards,

Marly A. Cardoso, Ph.D.

Academic Editor

PLOS ONE

Additional Editor Comments (optional):

Reviewers' comments:

Reviewer's Responses to Questions

**Comments to the Author**

1. If the authors have adequately addressed your comments raised in a previous round of review and you feel that this manuscript is now acceptable for publication, you may indicate that here to bypass the “Comments to the Author” section, enter your conflict of interest statement in the “Confidential to Editor” section, and submit your "Accept" recommendation.

Reviewer #1: All comments have been addressed

Reviewer #2: All comments have been addressed

Reviewer #3: All comments have been addressed

2. Is the manuscript technically sound, and do the data support the conclusions?

Reviewer #1: Yes

Reviewer #2: Yes

Reviewer #3: (No Response)

3. Has the statistical analysis been performed appropriately and rigorously? 

Reviewer #1: Yes

Reviewer #2: Yes

Reviewer #3: (No Response)

4. Have the authors made all data underlying the findings in their manuscript fully available?

Reviewer #1: Yes

Reviewer #2: Yes

Reviewer #3: (No Response)

5. Is the manuscript presented in an intelligible fashion and written in standard English?

Reviewer #1: Yes

Reviewer #2: Yes

Reviewer #3: (No Response)

6. Review Comments to the Author

Reviewer #1: Thank you for all the responses for each one of the comments. Although the study has some limitations, as already stated by the authors, there was a significant improvement of the manuscript. Now, the information is better described, contributing to the reader's interpretation of the study.

Reviewer #2: The authors made several improvements in the text. The points I raised in response to the initial submission have been sufficiently addressed. I consider that the article is suitable for publication in this Journal.

Reviewer #3: (No Response)

7. PLOS authors have the option to publish the peer review history of their article (what does this mean?). If published, this will include your full peer review and any attached files.

Reviewer #1: **Yes: **Isabel Giacomini

Reviewer #2: No

Reviewer #3: No

---

## [Editor Report · Acceptance letter]

22 Feb 2021

PONE-D-20-34834R1 

Relationship between exclusive breastfeeding and brain-derived neurotrophic factor in children 

Dear Dr. Solera-Martínez:

I'm pleased to inform you that your manuscript has been deemed suitable for publication in PLOS ONE. Congratulations! Your manuscript is now with our production department. 

Kind regards, 

on behalf of

Dr. Marly A. Cardoso 

Academic Editor

PLOS ONE